# Immune Suppression Induced by *Gallibacterium anatis* GtxA During Interaction with Chicken Macrophage-Like HD11 Cells

**DOI:** 10.3390/toxins12090536

**Published:** 2020-08-20

**Authors:** Bo Tang, Anders M. Bojesen

**Affiliations:** Department of Veterinary and Animal Sciences, Faculty of Health and Medical Sciences, University of Copenhagen, Stigboejlen 4, DK-1870 Frederiksberg C, Denmark; tangbo198908@163.com

**Keywords:** macrophage, invasion, *Gallibacterium anatis*, intracellular survival, ∆*gtxA* mutant

## Abstract

The RTX toxin GtxA expressed by *Gallibacterium anatis* biovar *haemolytica* has been proposed a major virulence factor during disease manifestations in the natural host, the chicken. To better understand the role of GtxA in the pathogenesis of *G. anatis,* we compared the GtxA expressing wildtype strain with its isogenic ∆*gtxA* mutant that was unable to express GtxA during exposure to chicken macrophage-like HD11 cells. From adhesion and invasion assays, we showed that GtxA appears to promote adhesion and invasion of HD11 cells. By using quantitative RT-PCR, we also demonstrated that the *G. anatis* expressing GtxA induced a mainly anti-inflammatory (IL-10) host cell response as opposed to the pro-inflammatory (IL-1β, IL-6 and TNF-α) response induced by the GtxA deletion mutant. Interestingly, these results, at least partly, resemble recent responses observed from spleen tissue of chickens infected with the same two bacterial strains. The effect of the GtxA toxin on the type of cell death was less clear. While GtxA clearly induced cell death, our efforts to characterize whether this was due to primarily necrosis or apoptosis through expression analysis of a broad range of apoptosis genes did not reveal clear answers.

## 1. Introduction

*Gallibacterium anatis* (*G. anatis*) is a genus in the *Pasteurellaceae* family [1,2]. The haemolytic biovar of *G. anatis* [3,4] has been recognized as a common part of the microbiota of the upper respiratory tract and the lower genital tract in healthy chickens [5,6,7]. The bacterium has, however, also been proposed as a major cause of infectious salpingitis and peritonitis in egg-laying chickens [8,9,10], where it leads to decreased egg production, increased mortality and lowered animal welfare. Recently, significant progress has been made in the knowledge of *G. anatis* epidemiology and pathology in its natural host, the chicken [7,11]. However, factors associated with the interaction with host immune cells like avian macrophages remain poorly understood [12].

A crucial step in bacteria–host interactions is attachment to the host surfaces allowing colonization [11,12]. For *G. anatis,* colonization of mucosal surfaces may go undetected by the host immune system [13]. Yet, the interactions may activate a complex cascade of molecular crosstalk at the host–pathogen interface and lead to a diverse set of downstream processes, including modulation of the innate host defenses [11,12,14,15]. Here, avian macrophages play a significant role in the initial defense against microbial infections [16,17,18].

*Gallibacterium anatis* produces several virulence factors, including cytotoxins like the RTX-like toxin GtxA [19,20], fimbriae [21], outer membrane vesicles [22], capsule [23] and other virulence determinants [12]. RTX toxins, like GtxA, have the ability to induce pores in the plasma membrane of host cells, which eventually can lead to necrosis or apoptosis of the target cells [24,25]. The related RTX toxin HlyA, which is secreted by pathogenic variants of *E. coli,* can promote host cell death via lysis, necrosis or apoptosis, depending on the target host cell, toxin concentration and duration of exposure [25,26].

GtxA shares several structural features with HlyA [24,25], and has been shown to induce a strong leukotoxic effect in vitro when exposed to avian macrophage-like HD-11 cells [27]. Induction of apoptosis and necrosis by *G. anatis* has also been demonstrated in primary chicken oviduct epithelial cells by Zhang and co-workers [28]. From in vivo trials in chickens, we recently reported that a GtxA expressing *G. anatis* strain (12656-12) induced increased levels of pro-inflammatory cytokines in ovarian tissue and a primarily anti-inflammatory response in spleen tissue, when compared to its GtxA devoid mutant. Here, an increased fraction of apoptopic cells was present in chickens infected with the GtxA expressing *G. anatis* wildtype compared to its isogenic GtxA mutant [20]. Based on that, GtxA appeared to be an important factor in the pathogenesis of *G. anatis* during infection of its natural host. To understand better the role of GtxA at the subcellular level, the aim of the present investigation was to study specific steps in the pathogenesis when chicken macrophage-like HD11 cells were exposed to a GtxA expressing *G. anatis* strain and its isogenic gtxA deletion mutant *(*∆*gtxA*), respectively. Specifically, we aimed at investigating the molecular events taking place during the early phases of attachment and invasion of avian host immune cells.

## 2. Results

### 2.1. Establishment of an Appropriate Multiplicity of Infection (MOI)

To establish a useful MOI, a preliminary adherence assay with *G. anatis* 12656-12 was done using the following MOIs: 10:1 (0.7 × 10^7^ CFU/mL), 50:1 (3.5 × 10^7^ CFU/mL), and 100:1 (0.7 × 10^8^ CFU/mL) bacteria per cell, respectively. During the initial 60 min, there was a positive correlation between adherence and time (Figure 1). At all MOIs tested, the apparent adhesion capacity was approx. 3–4 fold higher for MOI 10: 1 compared to MOI 50: 1 and 100: 1 (*p* ≤ 0.01) in three independent assays, respectively. The adhesion-level was overall highest after incubation for 60 min for MOI 10: 1 (*p* ≤ 0.05). In the subsequent assays, a MOI of 10:1 was used for all bacterial strains.

### 2.2. Bacterial Invasion and Intracellular Survival

The ability of the ∆*gtxA* mutant to enter HD11 cells was analyzed using a gentamicin protection assay to determine the invasion and intracellular capacity. The results showed significantly lower invasion ratio (0.09 to 0.13) of ∆*gtxA* mutant compared to the parent strain *G. anatis* 12656-12 (*p* ≤ 0.05) (Figure 2). The invasion ratios of the ∆*gtxA* mutant were nearly two-fold lower than that of WT bacteria on pre-fixed HD11 cells (*p* ≤ 0.01), indicating partial impairment of invasion of the ∆*gtxA* mutant at least during the first six hours of exposure.

To determine whether the expression of *gtxA* was required for invasion and intracellular survival, we compared the number of CFU of *G. anatis* WT and its isogenic ∆*gtxA* mutant strain recovered at 2, 4, 6, and 24 h PI, respectively (Figure 2). At four and six hrs PI, the number of WT bacteria invading the HD11 monolayer was significantly increased compared to the ∆*gtxA* mutant strain (*p* < 0.001). At 2 h PI, there was no difference found between the two strains. The number of intracellular *G. anatis* wild-type bacteria seemed to peak at 4 h and 6 h PI and then sharply decline. No *G. anatis* WT cells were detected intracellularly at 24 h PI.

### 2.3. Bacterial Growth

Bacterial growth according to time was compared between *G. anatis* WT and ∆*gtxA*. Both *G. anatis* WT and the ∆*gtxA* mutant reached exponential and stationary phases at 2 h and 6 h after initiation, respectively (Figure 3). There was no significant difference between the growth curves of the *G. anatis* WT and ∆*gtxA* mutant strains, suggesting that deletion of GtxA does not influence significantly the growth of *G. anatis* (Figure 3).

### 2.4. Viability of HD11 Cells Following Bacterial Exposure

The fraction of viable HD11 cells was measured following exposure to *G. anatis* WT and the ∆*gtxA* mutant, respectively. There was a significant reduction in the percentage of viable cells exposed to the WT and mutant, respectively (Figure 4). At 2 h PI, 97.9% and 78.1% viability were detected following infection with *G. anatis* ∆*gtxA* and WT, respectively. This was reduced to about 91.3% and 66.7% after 4 hrs incubation with *G. anatis* ∆*gtxA* and WT, respectively.

### 2.5. Real-Time Relative Quantification of Cytokine Expressions

The qRT-PCR analysis was performed to determine which cytokine genes were expressed in HD11 cells following exposure to the *G. anatis* Δ*gtxA* mutant and the parent strain. The cytokines were chosen to examine the difference between the two types of strains in their ability to elicit an immune response. The results of IL-1β, IL-6, IL-10, and TNF-α mRNA expression in HD11 cells infected with two different strains are displayed in Figure 5. The expression of IL-1β and IL-6 mRNA was significantly increased in HD11 cells infected with *G. anatis* Δ*gtxA* mutant, compared to the *G. anatis* WT at 2 PI. Moreover, the IL-10 mRNA expression was significantly down-regulated in HD11 cells treated with the *G. anatis* Δ*gtxA* mutant, compared with the WT at 2 PI. At 2 h PI no significant difference was observed in the TNF-α mRNA expression between the two bacterial strains. However, the TNF-α expression increased significantly in HD11 cells exposed to the ∆*gtxA* mutant, compared to the WT at 6 h PI. These results indicate that GtxA expression suppressed the induction of IL-1β, IL-6 and TNF-α, and induced expression of IL-10 in HD11 cells during the initial phases of exposure.

### 2.6. Real Time Relative Quantification of Apoptosis Gene Expression

To understand the reasons for the higher survival capacity of the ΔgtxA mutant in HD11 cells, and to study the role of GtxA in the pathogenesis of avian pathogenic G. anatis in vitro, we used qRT-PCR to investigate the expression of selected apoptosis genes including caspase-3, -8, -9 and the pro-apoptosis *bax* or anti-apoptosis *bcl-2* genes in chicken HD11 cells (Figure 6). Expression of caspase-3 and *bax* was down-regulated at 2 and particularly at 6 h PI in both the Δ*gtxA* and the WT group compared with control group, respectively (Figure 6). At six hours PI, the expression of caspase-8 appeared down-regulated in both the Δ*gtxA* and the WT group, compared to the control. Caspase-9 was initially up-regulated in the WT group only but down-regulated in the same group at 6 h PI. *Bcl-2* was down-regulated in the Δ*gtxA* mutant at 2 h, whereas both the mutant and the WT exposed cells expressed a five-fold lower level of *bcl-2* at 6 h compared to the control.

### 2.7. Flow Cytometric Analysis of HD11 Cell Death

The proportion of live and dead cells exposed to WT and Δ*gtxA* strains was investigated by flow cytometry using annexin VFITC/Pi double staining at 2 h and 6 h PI, respectively. As shown in Figure 7A, at 2 h PI, the fraction of live HD11 cells was lower in the WT exposed cells (70.4%) than in the Δ*gtxA* exposed cells (79.8%) and the control group (80.6%), respectively. These results clearly indicate that GtxA has the ability to induce *G. anatis*-induced cell death in HD11 cells. At 6 h PI the fraction of dead cells exposed to *G. anatis* WT had increased further to 36.2%, compared with 16.8% in the non-infected control group. However, in the Δ*gtxA* exposed group the fraction of death cells was 22.1% (Figure 7B), which further supported that the deletion of *gtxA* lowered the *G. anatis*-induced cell death.

## 3. Discussion

Our aim was to characterize the role of GtxA during the initial processes of the pathogenesis. In our assays aiming at adherence and invasion, the highest adherence capacity of the *G. anatis* wildtype was reached at 60 min PI, which could suggest that the corresponding receptors on the HD11 cell surfaces got saturated thereafter. This result was consistent with previously published results from use of primary chicken oviduct epithelial cells (PCOECs), although the peak of adherence in that investigation was reached slightly earlier [28]. Several adhesins, such as F17-like fimbria, capsular polysaccharides and secreted surface proteins, have been suggested as being involved in the adherence process [22,29,30,31,32,33,34]. The number of adhered ∆*gtxA* mutant bacteria was nearly two-fold lower than that of *G. anatis* WT on pre-fixed HD11 cells (Figure 2). This indicates that GtxA is involved in the attachment of HD11 cells, and at least partly explains the initial impairment of virulence observed for the ∆*gtxA* mutant.

As described previously, the ∆*gtxA* mutant was severely attenuated in experimentally infected chickens [27]. In the current investigation, we found lower HD11 intracellular counts of the ∆*gtxA* mutant strain at 4, 6 h PI compared to the *G. anatis* WT. Subsequently, a higher relative number of intracellular ∆*gtxA* mutant bacteria were counted as practically no WT bacterial cells were recovered after 6 h PI (Figure 2). These results indicate that the WT adhered and invaded at a high level during the initial phase of exposure. Later, the infected HD11 cells may have suffered from the toxic effect of the GtxA toxin, which seemed to leave a large fraction of the exposed HD11 cells apoptotic and/or necrotic (Figure 6). Disintegration of the HD11 cell may have exposed the intracellularly-based *G. anatis* to the gentamicin in the surrounding medium and explain the low number of viable bacteria and HD11 cells later than 6 h PI.

In a recent investigation, a highly virulent *G. anatis* strain was shown to induce production of pro-inflammatory cytokines during exposure to primary chicken oviduct epithelial cells [28]. In the current study, we compared the cytokine mRNA expression levels in HD11 cells infected with the *G. anatis* WT and ∆*gtxA* mutant using qRT-PCR. The previous report suggested an increased expression of IL-6, TNF-α, and IFN-γ by the PCOECs, indicating induction of a pro-inflammatory reaction [28]. We investigated if GtxA had a primarily pro-inflammatory (IL-1β, IL-6) or anti-inflammatory (IL-10) response, respectively, following exposure to HD11 cells. Our data showed that the expression of IL-1β was highly up-regulated while the expression of IL-10 was down-regulated in HD11 cells exposed to the *G. anatis* ∆*gtxA* mutant group. These results are, at least partly, consistent with the results from our previously reported in vivo investigation where the expression of IL-10 was up-regulated in the spleen while the TNF-α expression was suppressed following infection with the GtxA expressing WT [20]. TNF-α is generally considered to have an inverse relation to IL-10, where IL-10 promotes macrophage deactivation as opposed to TNF-α, which usually is regarded pro-inflammatory and macrophage activating. The effect of GtxA thus appeared to dampen the pro-inflammatory host reaction. The somewhat similar results obtained from HD11 cells (in vitro) and spleen tissue (in vivo) indicate that the HD11 cells may be used as a biologically relevant proxy prior to investigations of virulence factors or immune-activating (e.g., vaccine protoypes) in animals and support the 3R (replace/reduce/refine) animal protection initiative.

We hypothesized that one of the mechanisms by which *G. anatis* causes oophoritis and upper respiratory tract lesions may be through induction of apoptosis and necrosis of affected host cells. We aimed to test if the observed inflammatory response and cell death generated by HD11 cell was dependent on GtxA expression. The ability of bacterial pathogens to promote or inhibit apoptosis in eukaryotic cells is an emerging theme in the study of bacterial pathogenesis [35]. Several approaches enable bacterial pathogens of replicating in host cells while preventing apoptosis by repairing the mitochondrial perturbations, destroying cytochrome c release or preventing caspase activation [35].

Although GtxA has been shown to be vital for *G. anatis*-induced leukotoxicity and a major virulence factor during infections in chickens [20], several questions regarding the specific roles of GtxA remain. RTX toxins, including the *Escherichia coli α-hemolysin* (HlyA), *Actinobacillus pleuropneumoniae* (ApxI A), *Actinobacillus actinomycetemcomitans* (LtxA) and *Mannheimia haemolytica* (LktA) [24], have all been shown to induce apoptosis and cell death in a host-specific manner. Generally, RTX toxins bind to β2-integrins and induce changes associated with necrosis and cell lysis at high toxin concentrations, whereas cells exposed to lower toxin concentrations trigger signalling cascades promoting apoptosis-like cell death [36]. The molecular mechanisms leading to RTX-toxin cell death are complex and appear to affect cell homeostasis by destructing the host cell membrane integrity through formation and fusion of trans-membrane pores leading to cytolysis at high toxin concentrations [25]. At lower concentrations the toxin-mediated perturbations can either promote short-term activation of signaling cascades, resulting in rapid apoptosis-like cell death or programmed cell death destroying host cell integrity during a course of days [25,37,38,39]. Receptor-mediated activation of caspase-8 and mitochondrial activation of caspase-9 represent two common pathways at induction of apoptosis [38,40,41]. Subsequent effector caspases, like caspase-3, -6 and -7, are likewise hallmarks of apoptotic cell death [40].

In the current work, we found that HD11 cells expressed a significantly lower level of caspase-3 at 2 h PI following exposure to GtxA expressing *G. anatis,* while caspase-9 was up-regulated. At 6 h PI, a highly significant decrease in the expression of caspase-9 was observed, indicating that GtxA induced apoptosis is regulated in a caspase-9 dependent manner [41]. *G. anatis* thus appears to have the ability to escape antimicrobial activity asserted by HD11 cells, at least partly, by preventing caspase activation at different points along the apoptotic pathway. These findings are somewhat consistent with previous reports on *S. flexneri*, which might block apoptosis by targeting the already activated form of caspase-9 or the inactive form of caspase-3 thereby preventing caspase-3 activation [35]. We also characterized the expression of the anti-apoptosis or pro-survival *bcl-2* gene, which was dramatically down-regulated in the Δ*gtxA* exposed group, compared to the WT at 2 h PI. At 6 h PI, both the WT and the Δ*gtxA* exposed groups expressed significantly lower amounts of *bcl-2* mRNA than the control group. Again, these results indicate that GtxA may actively prevent host cell destruction by promoting a pro-survival response during the early phases of host cell exposure to GtxA.

## 4. Conclusions

We aimed at providing further insight into the role of GtxA during interaction with the avian immune system, here represented by the macrophage-like HD11 cell line. Although several pieces in the puzzle are still lacking, we believe the first contours of the pathogenesis, including both bacterial and host factors, respectively, have become apparent. While GtxA clearly has the ability to cause host cell lysis, GtxA also appears able of dampening the inflammatory host response based on an initial over-expression of IL-10 and a corresponding low-level expression of TNF-α. On the contrary, the Δ*gtxA* mutant induced a clear pro-inflammatory response, here represented by a very high expression of IL-1β, IL-6 and TNF-α. Furthermore, although less clear, GtxA initially seemed to promote partial host cell survival through an anti-apoptotic *bcl-2* response, while at a later stage that was less apparent. In conclusion, GtxA appears to be a main factor to control in order to mitigate the negative effects of a *G. anatis* biovar *haemolytica* on the host immune system and thus remain an obvious vaccine target.

## 5. Materials and Methods

### 5.1. Bacterial Strains and Growth Condition

The wild-type strain *Gallibacterium anatis* 12656-12 (*G. anatis* 12656-12) is a gentamicin-susceptible (MIC <1 mg/L) haemolytica strain derived from the liver of a chicken with septicaemia [7]. Avian pathogenic strain *G. anatis* 12656-12 wild-type (WT) and *G. anatis* ∆*gtxA* mutant were cultivated on brain heart infusion (BHI) (Oxoid, Basingstoke, UK) agar supplemented with 5% citrated bovine blood in a closed plastic bag [27]. Single colonies were incubated on BHI in an orbital shaking incubator overnight at 37 °C. The overnight cultures were transferred to a fresh BHI and cultured for 2 h to reach an exponential growth phase of a highly invasive phenotype. Prior to use, the bacteria were pelleted by centrifugation, resuspended in phosphate buffered saline (PBS) to OD_600_ = 0.2 and adjusted at a final concentration of approximately 2.5 × 10^8^ colony-forming unit (CFU)/mL. The bacterial concentration in each inoculum was verified by plate counts on BHI plates in duplicate.

### 5.2. Cell Lines and Culture Conditions

The MC29 bone marrow-transformed chicken macrophage-like cell line HD11 [42] was maintained in Roswell Park Memorial Institute (RPMI) 1640 medium added GlutaMAXTM-I, 25 mM HEPES (Gibco, Carlsbad, CA, USA), 2.5% chicken serum, 10% heat-inactivated foetal bovine serum (ThermoFischer, Gibco, South America) and 25μg/mL gentamicin. The HD11 cells were incubated at 37 °C in an atmosphere of 5% CO_2_. Cell concentrations were adjusted, and aliquots of cells (4 × 10^6^ cells/mL) were kept on ice until use. Prior exposure to bacteria, the culture medium of each sample was replaced with RPMI without antibiotics. The HD11 cell suspension was seeded into each well at 100 μL/well for 96-well plates and 600 μL/well for 24-well plates and allowed to grow to approximately 85% confluence before used for assays. The 96-well plates were used for multiplicity of infection (MOI) assay, whereas the 24-well plates were used for the invasion assays. For the qRT-PCR and apoptosis, assays cells were seeded at 4 × 10^8^cells/mL in 12-well plates. The entire experiment was performed in triplicate on two independent occasions.

### 5.3. Establishment of an Appropriate Multiplicity of Infection (MOI)

Preliminary MOI assays were performed to assess the infectivity of *G. anatis* in HD11 cells and determine an optimal concentration range of bacteria to be used in the subsequent adhesion and invasion assays. Incubation of *G. anatis* 12656-12 (WT and ∆*gtxA*) was performed in BHI to reach a final concentration of approximately 2.5 × 10^8^ CFU/mL. Subsequently, the bacterial preparations were washed three times with PBS and resuspended in RPMI culture medium without antibiotics. After 1h incubation period, monolayers of HD11 cells incubated in 24-well plates were added to bacterial suspensions of three different MOIs: 10:1 (0.7 × 10^7^ CFU/ mL), 50:1 (3.5 × 10^7^ CFU/mL) and 100:1 (0.7 × 10^8^ CFU/mL) bacteria per HD11-cell [28]. Subsequently, at 30, 60 and 90 min, samples of 100 mL infected HD11 cells were washed three times in pre-warmed PBS to remove the non-adherent bacteria. The remainder of HD-11 cells were incubated with RPMI culture medium containing gentamicin (100 mg/mL) for one hour, and subsequently cells were lysed with 1 mL of 0.1% Triton X-100 solution (Sigma, Søborg, Denmark) [43]. After homogenization, a 100 mL aliquot of lysed cell suspension was diluted into 900 mL PBS enabling 10-fold serial dilutions to be inoculated on BHI ager plates for quantification of bacterial colonies. The number of colony-forming units (CFUs) after 24 h of incubation was determined in triplicate.

### 5.4. Cell Invasion and Intracellular Survival Assay

Following optimization of the HD11 and *G. anatis,* co-culture 24-well cell culture plates containing approximately 0.7 × 10^6^ HD11 cells/well were incubated for 1 h prior to infection. Bacterial suspensions (2.5 × 10^8^ CFU/mL) were prepared as described above and inoculated for 1h into each well at 37 °C in a 5% CO_2_ atmosphere. To quantify the invading bacteria, extracellular bacteria were removed by washing the HD11 cells three times and subsequently adding RPMI medium containing gentamicin (100 mg/mL). The cells were then left at 37 °C for 1 h. Following gentamicin treatment, the HD11 monolayers were washed with PBS and the cells were lysed by addition of 1 mL 0.1% Triton X-100 for 10 min to release intracellular bacteria. Serial dilutions (1:10) of each well content were then inoculated on BHI agar plates to allow CFU determination after 24 h of incubation at 37 °C. The number of invading bacteria was assessed at 1, 2, 4, 6, 8, 12 and 24 h PI, respectively. The invasion ratio was calculated as follows: number of invaded bacteria/the initial bacterial inoculation number. The ability of bacterial to survive in HD11 cells was related to the ratio of intracellular bacteria, which was determined by dividing the number of intracellular bacteria by the initial invasion of bacterial number [17,28].

### 5.5. Trypan Blue Exclusion Assay

Trypan blue viability measurement was performed by standard method [14,44,45]. The 24-well plates, containing 0.7 × 10^6^ cell/well, were infected with *G. anatis* 12656-12 as invasion assay described above. Cellular counts were done after incubation with fresh culture media supplemented with gentamicin (100 mg/mL) after 1, 2, 4, 6, 8, 12 or 24 h, respectively. After gentamicin treatment, infected cells were washed three times with PBS and lysed by trypsin–EDTA following incubation for 10 min. One hundred microliter cell suspensions were mixed with an equal volume of 0.4% trypan blue solution (Sigma). The suspension was loaded into a Neubauer hemocytometer and scored with an inverted light microscope (Helmut Hund GmbHD 6330; Wetzlar, Germany) at low magnification. The percentage of viable cells was determined by dividing the number of live infected cells by a number of live non-infected cells [14,44].

### 5.6. Growth Assay

Growth curve profiles were constructed to determine the significance of GtxA on *G. anatis* growth. The growth of the *G. anatis* WT and ∆*gtxA* mutant strains were compared by inoculating in BHI medium with shaking at 220 rpm and culturing at 37 °C for 12 h. Bacterial growth was estimated by OD_600_ performed three times at 1 h intervals.

### 5.7. Flow Cytometric Analysis of HD11 Cell Death

To identify the percentage of HD11 cells undergoing cell death in vitro, the Annexin V-FITC Apoptosis Detection Kit (BD Biosciences, San Jose, CA, USA) was used. Infection of HD11 cell cultures was performed with *G. anatis* WT and ∆*gtxA* during 2 h and 6 h, respectively. In parallel, an uninfected HD11 group served as the negative control. From either group in six-well tissue culture plates, material was harvested by adding 0.25% trypsin and washing three times with PBS. The cells were stained with 5 uL FITC-conjugated annexin V (Annexin V-FITC) and 3 uL Propidium iodide (Pi) in a volume of 100 mL on ice for 30 min. Subsequently, the mixture was incubated at room temperature for 20 min in the dark. The FITC and Pi fluorescence was measured by flow cytometry (BD Accuri™C6, Beckman Coulter, Indianapolis, IN, USA) within an hour. The fractions of live and dead HD-11 cells were quantified and compared two- and six-hours post infection, respectively.

### 5.8. Real-Time Quantitative RT-PCR

To assess the mRNA expression profile of the HD11 cells, samples were stored in RNALater solution (Qiagen, Hilden, Germany) at −80 °C prior to RNA purification. The six-well plates containing 4 × 10^8^cells/well were infected with *G. anatis* 12656-12 WT and ∆*gtxA* as previously described. In parallel, an uninfected HD11 group served as the negative control. Samples from each of the three groups were obtained after 2 h and 6 h, respectively, extracted and homogenized in 1 mL RTL buffer individually. Subsequent to homogenization and centrifugation for 1 min, the upper phase containing RNA was collected and purified using the RNeasy mini kit (Qiagen, Hilden, Germany) in accordance with the manufacturer’s instructions. The concentration and purity of RNA were determined by a Nanodrop 1000 spectrophotometer (Thermo Scientific, Wilmington, DE, USA). Five ug of RNA extracted from each sample were immediately reverse-transcribed into cDNA using M-MLV reverse transcriptase kit (Invitrogen) according to the manufactures protocol. An amount of cDNA corresponding to 20 ng of reverse-transcribed RNA was amplified by qRT-PCR, using specific primers obtained from the NCBI database and synthesized at Germany listed in Table 1 and Table 2. The cDNA product was used as template in 25 μL PCR reactions in 96-well microplates using FastStart Essential DNA Green Master Mix (Roche, Penzberg, Germany). The three-step amplification and signal detection were performed using a LightCycler R^®^ 96 (Roche) with an initial preincubation at 95 °C for 10 min followed by 40 cycles of 95 °C for 15 s, 56 °C for 30 s and 72 °C for 30 s, respectively. Apoptosis and cytokines mRNA expression levels in HD11 cells were quantified using qRT-PCR. β-actin expression was used as an internal control to normalize the quantification of target genes. The relative quantification of apoptosis gene-specific expression was calculated using the 2^−ΔΔCt^ method after normalization with chicken β-actin [46]. Quantification of cytokine gene expression of Il-6, IL- IL-1β, IL-10, TNF-α and β-actin genes was calculated using the same formula. All qRT-PCR reactions were performed in triplicate.

### 5.9. Statistical Analysis

All data were expressed as a mean and a standard error of the mean (SEM) from three independent experiments performed in triplicate. To compare the relative gene expression levels, we used the 2^−ΔΔc(t)^ method [46]. The statistical analyses were conducted using Student’s *t*-test in GraphPad Prism version 7 (GraphPad Software, San Diego, CA, USA). A *p*-value ≤ 0.05 was considered significant, while a *p*-value ≤ 0.01 and *p* ≤ 0.001 was considered moderate and highly significant, respectively.

## Figures and Tables

**Figure 1 toxins-12-00536-f001:**
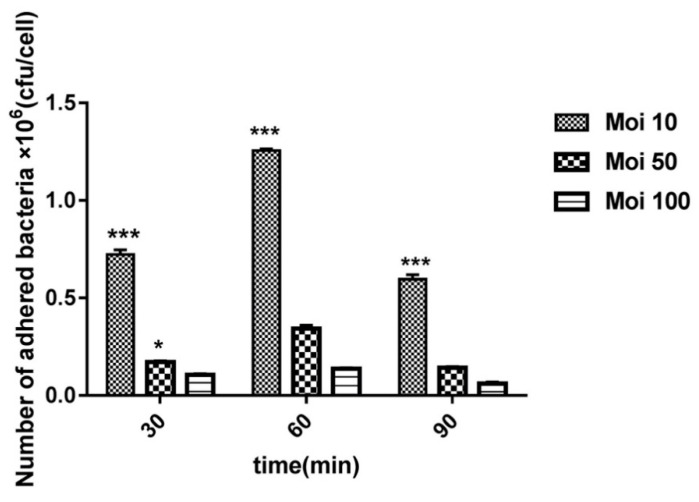
Kinetics of adherence of *G. anatis* 12656-12 strain to HD11 cells after 30, 60 and 90 min of incubation at three different multiplicities of infection (MOIs). Significant differences were labeled (*p* ≤ 0.05) * and (*p* ≤ 0.001) ***.

**Figure 2 toxins-12-00536-f002:**
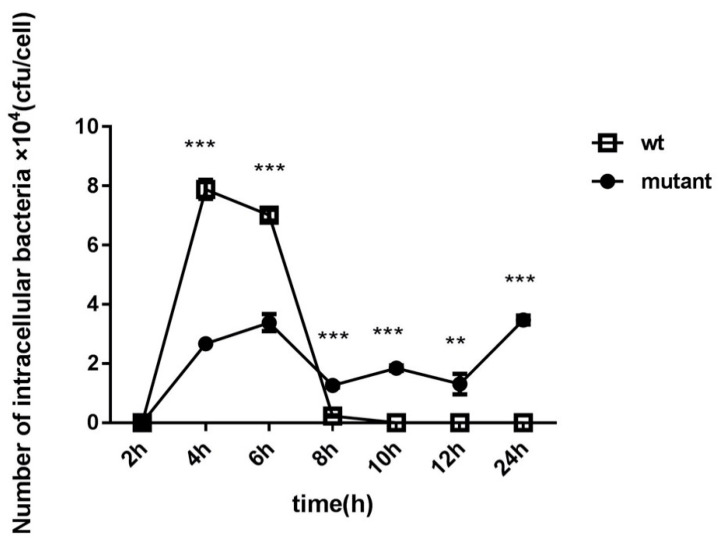
Gentamicin protection assay to determine intracellular survival ability of *G. anatis* 12656-12 and *G. anatis* ∆*gtxA* in the chicken HD11 cells over time. The ability of the *G. anatis* ∆*gtxA* mutant to enter and survive in the HD11 cells was compared to its *G. anatis* 12656-12 parent strain. Bacterial counts (CFU/cell) were obtained and compared from lysed HD11 cells at 2, 4, 6, 8, 10, 12 and 24 h following addition of gentamicin ** (*p* ≤ 0.01) *** (*p* ≤ 0.001).

**Figure 3 toxins-12-00536-f003:**
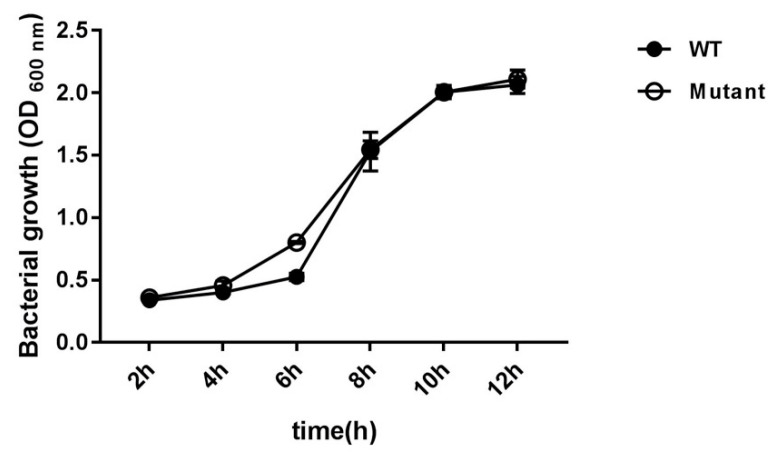
Growth curves of the *G. anatis* wild-type (WT) strain and the ∆*gtxA* mutant strain. Growth was assessed by measurement of the OD_600_ at 2, 4, 6, 8, 10 and 12 h. The graph is representative of three independent experiments.

**Figure 4 toxins-12-00536-f004:**
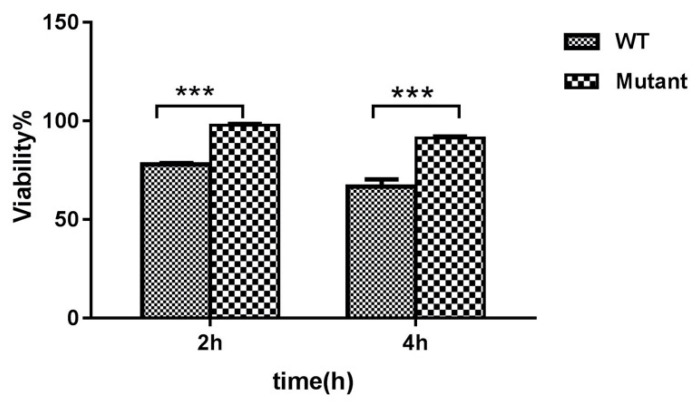
Effect of *G. anatis* infection on macrophage viability. HD11 cell viability assay following infection with the wild-type strain (*G. anatis* 12656-12) and *G. anatis* ∆*gtxA* mutant. Data shown represent means ± SEM from three independent experiments *** (*p* ≤ 0.001).

**Figure 5 toxins-12-00536-f005:**
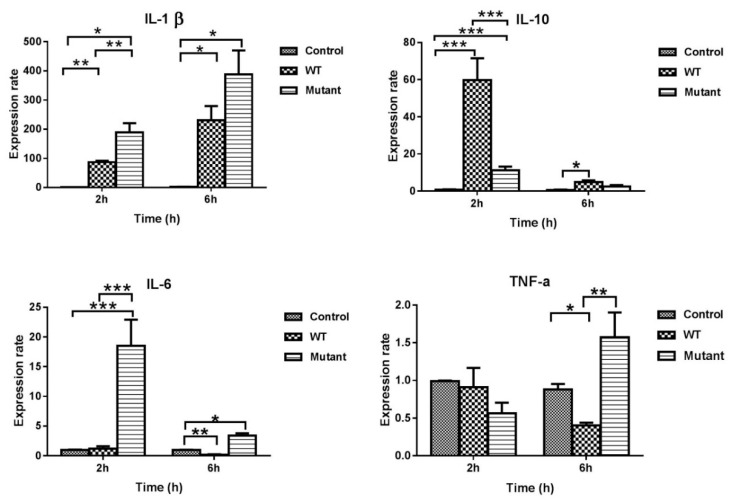
Real-time relative quantification of cytokine expressions. Cytokine expressions were measured on RNA isolated from HD11 cells in vitro. The relative gene expression levels were calculated as a ratio of stimulated WT and Δ*gtxA* to un-stimulated (non-infected control), respectively. Data were log2 transformed. Values shown are averages and SEM of three independent experiments. Asterisks indicate significance from the uninfected control cells (* *p* ≤ 0.05, ** *p* ≤ 0.01, *** *p* ≤ 0.001).

**Figure 6 toxins-12-00536-f006:**
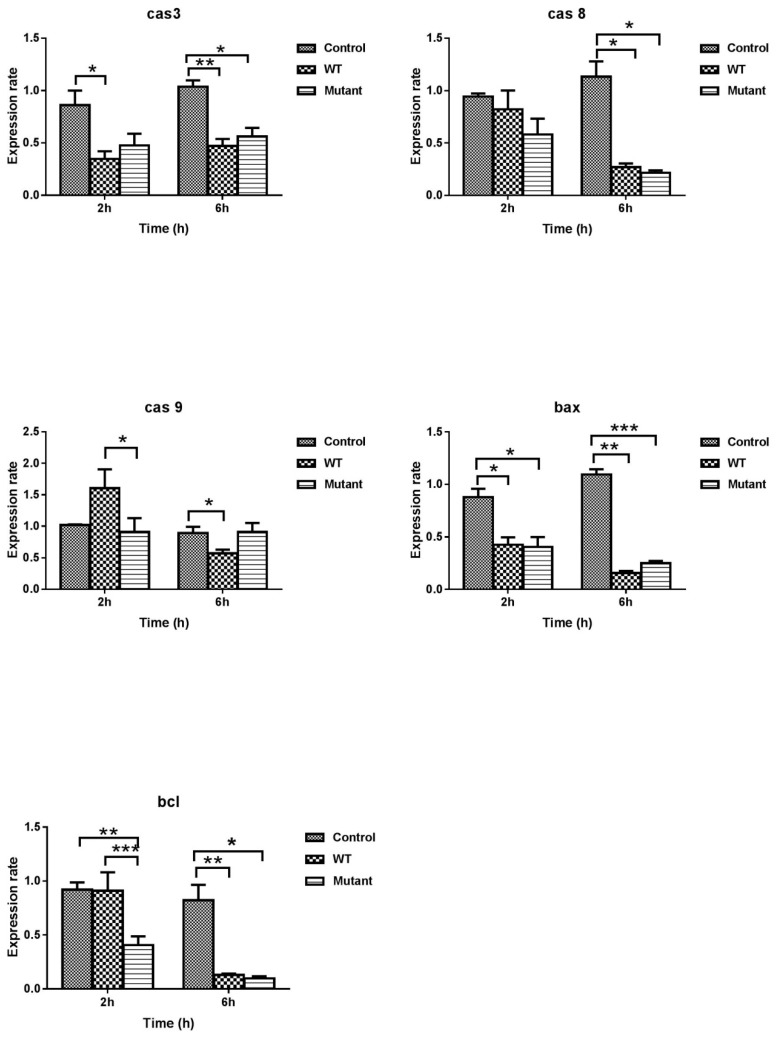
Real-time relative quantification of apoptosis gene expression. Apoptosis gene expression was measured on RNA isolated from HD11 cells in vitro. The relative gene expression levels were calculated as a ratio of stimulated WT and Δ*gtxA* to un-stimulated (non-infected control), respectively. Data were log2 transformed. Values shown are averages and SEM of three independent experiments. Asterisks indicate significance from the uninfected control cells (* *p* ≤ 0.05, ** *p* ≤ 0.01, and *** *p* ≤ 0.001).

**Figure 7 toxins-12-00536-f007:**
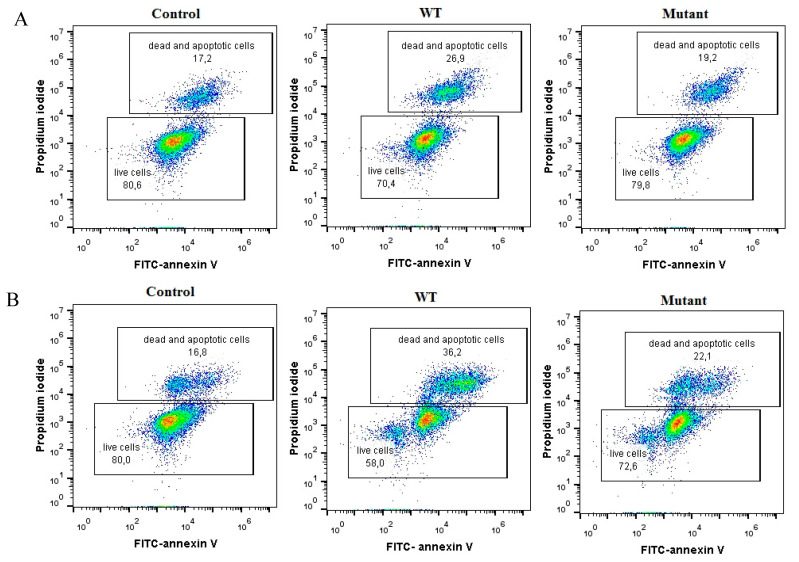
Effect of GtxA on apoptosis and necrosis in *G. anatis*-infected HD11 cells. The HD11 cells were stained with annexin V FITC/ Propidium iodide (Pi) and analyzed by flow cytometry. (**A**) Flow cytometric analysis of non-exposed HD11 cells compared to cells exposed to the WT or the ∆*gtxA* mutant strain at 2 h PI (**B**) Flow cytometric analysis of HD11 cells infected with WT and ∆*gtxA* mutant strain at 6 h PI. Cells were stained with FITC-Annexin V and Pi. The lower inserted box: the living cells and early apoptotic cells. The upper inserted box: late apoptotic cells and necrotic cells.

**Table 1 toxins-12-00536-t001:** List of primers used in qRT-PCR analysis of mRNA expression of the apoptotic proteins.

Name of Primer	Primer	Primer Sequence (5′–3′)	Accession	Conc. Used
Bcl-2	Forward	GATGACCGAGTACCTGAACC	NM205339	0.2 mM
	Reverse	CAGGAGAAATCGAACAAAGGC		
Bax	Forward	TCCTCATCGCCATGCTCAT	XM422067	0.4 mM
	Reverse	CCTTGGTCTGGAAGCAGAAGA		
Caspase-8	Forward	TGGCCCTCTTGAACTGAAAG	AY057940	0.4 mM
	Reverse	TCCACTGTCTGCTTCAATACC		
Caspase-9	Forward	CGAAGGAGCAAGCACGACAG	AY057940	0.2 mM
	Reverse	CCGCAGCCCTCATCTAGCAT		
Caspase-3	Forward	TGGCCCTCTTGAACTGAAAG	AY057940	0.4 mM
	Reverse	TCCACTGTCTGCTTCAATACC		
β-actin	Forward	TGCTGTGTTCCCATCTATCG	L08165	0.2 mM
	Reverse	TTGGTGACAATACCGTGTTCA		

**Table 2 toxins-12-00536-t002:** List of primers used in qRT-PCR analysis of mRNA expression of the cytokines.

Name of Primer	Primer	Primer Sequence(5′–3′)	Accession	Conc. Used
IL-6	Forward	GCTCGCCGGCTTCGA	AJ250838	0.2 mM
	Reverse	GGTAGGTCTGAAAGGCGAACAG		
IL-10	Forward	CATGCTGCTGGGCCTGAA	J621614	0.4 mM
	Reverse	CGTCTCCTTGATCTGCTTGATG		
β-actin	Forward	CCGCTCTATGAAGGCTACGC	L08165	0.4 mM
	Reverse	CTCTCGGCTGTGGTGGTGAA		
TNF-α	Forward	GCCCTTCCTGTAACCAGATG	NM_204267	0.2 mM
	Reverse	ACACGACAGCCAAGTCAACG

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
