# Peer review of "Immune Suppression Induced by Gallibacterium anatis GtxA During Interaction with Chicken Macrophage-Like HD11 Cells"

_toxins, 2020, doi:10.3390/toxins12090536_

Round 1
Reviewer 1 Report
The manuscript “Molecular events induced by Gallibacterium anatis 2 GtxA during interaction with chicken 3 macrophage-like HD11 cells” Toxins manuscript Nr.899302 is an original study on the immunological parameters induced by GtxA interaction of Gallibacterium anatis on chicken HD11 cells. The study brings up a new activity of the RTX toxin GtxA, the induction of IL10 and the resulting immune-suppression. This is of utmost importance in the role of GtxA in pathogenesis Gallibacterium anatis. Furthermore it is important for design of vaccines against G. anatis infections, indicating that vaccines should be devoid of GtxA. The research is carefully performed and well presented. It merits publication in ‘Toxins’.
The following points might improve the manuscript:
Title: I would make the title more attractive: Instead of “Molecular events induced by ….” I would write: “Immune suppression induced by …..”
Line 9: “gtxA” in italics
Line 12: “we” not “We”
Line 30: delete “despite these advances” (makes the sentence clumsy)
Line 35: change “activation” to “modulation”, since the result finally show immune-suppression.
Figures 5 and 6: Y-axis label “Fold change” is odd. I would suggest “expression rate” and explain in the figure legend “the expression rate is normalized to non-infected cells”.
Author Response
The manuscript “Molecular events induced by Gallibacterium anatis 2 GtxA during interaction with chicken 3 macrophage-like HD11 cells” Toxins manuscript Nr.899302 is an original study on the immunological parameters induced by GtxA interaction of Gallibacterium anatis on chicken HD11 cells. The study brings up a new activity of the RTX toxin GtxA, the induction of IL10 and the resulting immune-suppression. This is of utmost importance in the role of GtxA in pathogenesis Gallibacterium anatis. Furthermore it is important for design of vaccines against G. anatis infections, indicating that vaccines should be devoid of GtxA. The research is carefully performed and well presented. It merits publication in ‘Toxins’.
The following points might improve the manuscript:
Title: I would make the title more attractive: Instead of “Molecular events induced by ….” I would write: “Immune suppression induced by …..”
Good suggestion. The title has been changed to “Immune suppression induced by Gallibacterium anatis GtxA during interaction with chicken macrophage-like HD11 cells”
Line 9: “gtxA” in italics
The change has been made.
Line 12: “we” not “We”
The change has been made.
Line 30: delete “despite these advances” (makes the sentence clumsy)
Agree. The suggestion has been met.
Line 35: change “activation” to “modulation”, since the result finally show immune-suppression.
The suggested change has been made.
Figures 5 and 6: Y-axis label “Fold change” is odd. I would suggest “expression rate” and explain in the figure legend “the expression rate is normalized to non-infected cells”.
The label “Fold change” has been changed to “Expression rate”. The legend has also been rephrased.
Reviewer 2 Report
Manuscript ID toxins-899302
Review Report
In this paper the authors aim to clarify the role played by the RTX-like toxin GtxA in host pathogenesis using for that chicken macrophage-like HD11 cells and conducting different methods. They conclude that the GtxA toxin induces an anti-inflammatory response by overexpression of IL-10.
Major points
This referee would like to see an important control assay that is not shown here, a SDS-PAGE or WB to know which is the level of expression of the GtxA toxin at different time points. There are several conclussions that cannot be achieved without this control. Please include this figure as panel b in fig.3.
Do the authors have an idea of what factors may induce the pro-inflammatory response in the chicken macrophages incubated with the rtxA deletion mutant?
Line 351: Provide a reference for the 2 –ΔΔCt method, and describe it more extensively in the text.
Describe which is the control used in the cell death assay by flow cytometry (annexin V/PI)
Indicate in the text which has been the control group for the assay of quantification of apoptosis gene expression
Minor points
Line 8:” …we compared a the GtxA expresing 1 wild type…” a, or the, but not both
Line 9: In line 9 it has been used …gtxA deletion mutant…”, while in the key words it has been used ΔgtxA. Unify
Line 11: we showed that GtxA appear to promote; apprears
Line 12: “…We also demonstrated…” We in lowercase, correct it
Line 18: …expression analysis of a broad range of apoptosis did not reveal…; apoptosis genes or markers
Line 27: “…has been proposed a major cause of…”; has been proposed to be, has been proposed as
Lines 28, 30, 34
Line 163: “Cells were stained with annexin V FITC and pi.” In other parts of the text it has been used Pi. Unify
Line 190: “… induction of an pro-inflammatory…”. Correct “an”
Line 198: pro-inflammatory, line 199: proinflammatory. Unify
Line 201: “… through induction apoptosis and …”. Induction of
Line 322: …cwll…Correct it
Author Response
In this paper the authors aim to clarify the role played by the RTX-like toxin GtxA in host pathogenesis using for that chicken macrophage-like HD11 cells and conducting different methods. They conclude that the GtxA toxin induces an anti-inflammatory response by overexpression of IL-10.
Major points
This referee would like to see an important control assay that is not shown here, a SDS-PAGE or WB to know which is the level of expression of the GtxA toxin at different time points. There are several conclussions that cannot be achieved without this control. Please include this figure as panel b in fig.3.
We tend to disagree to this point. Even though it could be interesting to know about the expression levels of the GtxA toxin at different time points, this was really not the purpose of the experiment. Our aim was to show if the wildtype and the mutant had a comparable growth rate during a 12 hour period, which was the time period most of our assays were done within. Had there been a difference in the number of bacterial cell present that in itself could have influenced the outcome. This was not the case, which in our perception means that the difference observed between the wildtype and the mutant can solely be assigned presence or absence of GtxA.
We have previously investigated and published results on the expression of GtxA using the same strain and growth conditions as used in the current work (Kristensen et al., 2010; Kristensen et al., 2011). Briefly, GtxA is expressed in the wild type strain whereas the deletion mutant does not express GtxA at any time. We have shown variations in the amount of GtxA toxin in the culture supernatant yet our primary aim was not to compare the GtxA expression levels of wildtype between two time points but to compare the lesions inflicted by the wildtype and the mutant at different time points, respectively.
Kristensen BM, Frees D, Bojesen AM. GtxA from Gallibacterium anatis, a cytolytic RTX-toxin with a novel domain organisation. Vet Res. 2010;41(3):25. doi:10.1051/vetres/2009073
Kristensen BM, Frees D, Bojesen AM. Expression and secretion of the RTX-toxin GtxA among members of the genus Gallibacterium. Vet Microbiol. 2011;153(1-2):116-123. doi:10.1016/j.vetmic.2011.05.019
Do the authors have an idea of what factors may induce the pro-inflammatory response in the chicken macrophages incubated with the rtxA deletion mutant?
- anatis 12656-12 has been shown to express several additional immune-activating factors including fimbriae, outer membrane vesicles and capsule. This has previously been reported in the review Persson & Bojesen (2015).
Persson G, Bojesen AM. Bacterial determinants of importance in the virulence of Gallibacterium anatis in poultry. Vet Res. 2015;46(1):57. Published 2015 Jun 11. doi:10.1186/s13567-015-0206-z
Line 351: Provide a reference for the 2 –ΔΔCt method, and describe it more extensively in the text.
The reference for the 2 –ΔΔCt method (Livak and Schmittgen, 2001) has been has been added and the description has also been changes slightly.
Livak KJ, Schmittgen TD: Analysis of relative gene expression data using real-time quantitative PCR and the 2-ΔΔCt method. Methods. 2001, 25(4):402-408.
Describe which is the control used in the cell death assay by flow cytometry (annexin V/PI)
The control in the flow cytometry assays was HD11 unexposed to bacteria, so a negative control. This has been clarified further in the text (lines 327-328).
Indicate in the text which has been the control group for the assay of quantification of apoptosis gene expression.
As stated above the non-infected control group acted as the negative control in the flow cytometry assay. β-actin expression was used as an internal control to normalize the quantification of target genes. The relative quantification of apoptosis gene-specific expression was calculated using the 2−ΔΔCt method after normalization with chicken β-actin.
Minor points
Line 8:” …we compared a the GtxA expresing 1 wild type…” a, or the, but not both
Corrected.
Line 9: In line 9 it has been used …gtxA deletion mutant…”, while in the key words it has been used ΔgtxA. Unify
Corrected.
Line 11: we showed that GtxA appear to promote; apprears
Corrected.
Line 12: “…We also demonstrated…” We in lowercase, correct it
Corrected.
Line 18: …expression analysis of a broad range of apoptosis did not reveal…; apoptosis genes or markers
Corrected.
Line 27: “…has been proposed a major cause of…”; has been proposed to be, has been proposed as
Corrected.
Lines 28, 30, 34
We do not understand what the reviewer aims at here?
Line 163: “Cells were stained with annexin V FITC and pi.” In other parts of the text it has been used Pi. Unify
Corrected.
Line 190: “… induction of an pro-inflammatory…”. Correct “an”
Corrected.
Line 198: pro-inflammatory, line 199: proinflammatory. Unify
Corrected.
Line 201: “… through induction apoptosis and …”. Induction of
Corrected.
Line 322: …cwll…Correct it
Corrected.
Round 2
Reviewer 2 Report
Accept